# Anti-TNF Therapies Suppress Adipose Tissue Inflammation in Crohn’s Disease

**DOI:** 10.3390/ijms231911170

**Published:** 2022-09-22

**Authors:** Albert Boronat-Toscano, Diandra Monfort-Ferré, Margarita Menacho, Aleidis Caro, Ramon Bosch, Beatriz Espina, Francisco Algaba-Chueca, Alfonso Saera-Vila, Alicia Moliné, Marc Marti, Eloy Espin, Mónica Millan, Carolina Serena

**Affiliations:** 1Institut d’Investigació Sanitària Pere Virgili, Hospital Universitari Joan XXIII, Universitat Rovira i Virgili, 43007 Tarragona, Spain; 2Digestive Unit, Hospital Universitari Joan XXIII, 43007 Tarragona, Spain; 3Colorectal Surgery Unit, Hospital Universitari Joan XXIII, 43007 Tarragona, Spain; 4Department of Pathology, Oncological Pathology and Bioinformatics Research Group, Hospital de Tortosa Verge de la Cinta—IISPV, 43500 Tortosa, Spain; 5Sequentia Biotech, Carrer Comte D’Urgell 240, 08036 Barcelona, Spain; 6Colorectal Surgery Unit, General Surgery Service, Hospital Valle de Hebron, Universitat Autonoma de Barcelona, 08035 Barcelona, Spain; 7Colorectal Surgery Unit, General Surgery Service, Hospital La Fe, 46026 Valencia, Spain

**Keywords:** adipose tissue, infliximab, adalimumab, TNF, creeping fat

## Abstract

Anti-TNF biologics have been shown to markedly improve the quality of life for patients with Crohn’s disease (CD), yet one-third of patients fail to benefit from this treatment. Patients with CD develop a characteristic wrapping of visceral adipose tissue (VAT) in the inflamed intestinal area, termed creeping fat, and it is known that adipose tissue expansion influences the efficacy of anti-TNF drugs. We questioned whether anti-TNF therapies impact the creeping fat in CD, which might affect the outcome of the disease. Adipose tissue biopsies were obtained from a cohort of 14 patients with CD that received anti-TNF drugs and from 29 non-anti-TNF-treated patients (control group) matched by sex, age, and body mass index undergoing surgical interventions for symptomatic complications. We found that anti-TNF therapies restored adipose tissue morphology and suppressed immune cell infiltration in the creeping fat. Additionally, anti-TNF treatments appeared to markedly improve the pro-inflammatory phenotype of adipose-tissue macrophages and adipose-tissue-derived stem cells. Our study provides evidence that anti-TNF medications influence immune cells and progenitor cells in the creeping of patients with CD, suppressing inflammation. We propose that perilesional VAT should be considered when administering anti-TNF therapy in patients with CD.

## 1. Introduction

Crohn’s disease (CD) is a chronic inflammatory disease of the digestive tract that generally affects younger patients and is associated with significant morbidity and poor quality of life. While its etiology is not fully understood, genetic and environmental factors are likely to contribute together with changes in the intestinal microbiota (dysbiosis) with involvement of the intestinal barrier, all in a context of impaired immunity [1]. Indeed, CD is classically defined as a T-helper type 1 (Th1) immunological disorder that is characterized by trans-mural inflammation and the elevated production of tumor necrosis factor (TNF)-α, interleukin (IL)-12, and interferon (IFN)-γ.

A widely known but little-studied observation in patients with CD is the development of mesenteric fat that wraps around the inflamed segments of the intestine, also known as creeping fat. Perilesional visceral adipose tissue (VAT) expansion is a hallmark of CD and appears to be directly related to disease activity [2,3,4]. Creeping fat is associated with adipocyte hyperplasia, although it has been reported that adipocytes in this region are 75% smaller than equivalent cells from healthy subjects [5,6]. Adipose tissue hyperplasia results from the increase in adipocyte number via the recruitment and differentiation of adipose precursor cells, termed adipose tissue-derived mesenchymal stem cells or ASCs [7,8]. Intriguingly, post-surgical recurrence rates requiring intervention have been reported to decrease dramatically (from 40% to 2.9%) when the mesenteric adipose tissue is included during bowel resection [9,10], and a recent study has proposed the use of computed tomography-based assessment of mesenteric fat as a measure of intestinal fibrostenosis severity in patients with CD [4].

Anti-tumor necrosis factor (anti-TNF) therapies (including infliximab (IFX) or adalimumab (ADA)) have revolutionized the treatment of CD. Their use circumvents the need for steroid therapy, promotes mucosal healing, reduces hospitalization and surgeries, and, accordingly, considerably improves the quality of life for patients with this disease [11]. Nevertheless, one-third of patients fail to respond to anti-TNF therapy [12,13,14]. While many studies have examined the impact of anti-TNF therapies on the intestinal mucosa in CD [15,16,17,18], no studies to our knowledge have assessed the effect of these therapies on mesenteric creeping fat despite the fact that adipose tissue expansion is known to influence the efficacy of anti-TNF drugs. This is likely because of the dearth of animal models to investigate perilesional VAT in the context of CD. It would seem prudent to examine how anti-TNF therapies might impact perilesional adipose tissue and whether this contributes to disease amelioration. We hypothesized that anti-TNF therapies suppress inflammation in the perilesional adipose tissue. We investigated the effects of anti-TNF treatments on this tissue and its cellular subtypes.

## 2. Results

### 2.1. Adipose Tissue Is Remodeled by Anti-TNF Therapy, Which Restores Adipocyte Morphology and Suppresses Immune Cell Infiltration

The demographic and clinical characteristics of the patients with CD included in the present study are listed in Table 1. In total, 29/43 patients were non-anti-TNF-treated patients (control group) and 14/43 patients were receiving anti-TNF therapy (n = 9 ADA and n = 5 IFX). C-reactive protein was significantly higher in the control group compared to the anti-TNF treated group. To note, IL-6 levels were significantly higher in the control group compared to the anti-TNF-treated group (Table 1). An unpublished observation made by our surgical team noted that the in situ appearance of the perilesional VAT in patients undergoing surgery differed depending on whether or not they had previously received biological anti-TNF medication. We performed haematoxylin and eosin (H&E) staining on samples of the creeping fat of patients in both groups, observing that adipocytes of the former group were larger and more similar in size to those in the VAT of healthy individuals (Figure 1A, left panel) [19]. To corroborate these findings, we quantitatively measured adipocyte area with the Adiposoft plug-in for Fiji (ImageJ, NIH). The results indicated that the adipocyte area in the perilesional VAT was significantly greater in anti-TNF-treated patients than in the control group (Figure 1B). Furthermore, a significantly higher percentage of large adipocytes (>3000 µm^2^) and a significantly lower percentage of small adipocytes (<3000 µm^2^) were detected in the VAT of anti-TNF-treated patients (Figure 1C).

A typical feature of the creeping fat is immune cell infiltration. To examine whether this feature was influenced by anti-TNF therapy, we performed immunohistochemistry staining on perilesional VAT sections of all patients. The results revealed the presence of CD68-positive macrophages in sections from both groups (Figure 1A, central panel), and quantitative analysis revealed a significantly lower number of macrophages in sections from anti-TNF-treated patients (Figure 1D). Corroborating this finding, Western blot analysis of the monocyte/macrophage marker CD14 in creeping fat protein extracts revealed a trend for lower expression in anti-TNF-treated patients (Figure 1E), likely indicating a diminished infiltration of the monocyte/macrophage lineage in this tissue [20]. The results of double staining for CD68 and CD3 (specific for T lymphocytes) not only validated the lower macrophage infiltration in this tissue in anti-TNF-treated patients, but also indicated the presence of T lymphocytes (Figure 1A, right panel). Intriguingly, a significantly lower infiltration of T lymphocytes was detected in the creeping fat of anti-TNF-treated patients than of control patients (Figure 1F). Finally, the ratio of macrophages to T lymphocytes was slightly greater in the creeping fat of control patients than in anti-TNF-treated peers (Figure 1G).

### 2.2. Anti-TNF Therapies Influence the Creeping Fat, Reducing Inflammation in Various Cell Subtypes

Given our finding of a lower infiltration of immune cells in the creeping fat of anti-TNF-treated patients, we next aimed to elucidate whether this treatment also affected the immunological profile of macrophages and whether it has effects on VAT per se. We first analyzed the relative gene expression of pro-inflammatory and antigen-presentation markers using quantitative PCR (qPCR) in macrophages isolated from the creeping fat of both groups of patients. The results showed than the expression of the pro-inflammatory markers *TNFA*, *IL6*, and *IL1B* was significantly lower in macrophages isolated from anti-TNF-treated patients than in control patients (Figure 2A). A similar result was found for the expression of the antigen-presentation markers *HLA-DM*, *HLA-DPB*, and *CD40* (Figure 2B). We expanded this analysis to determine whether anti-TNF therapy impacted the inflammatory status of the perilesional VAT per se. The results showed that anti-TNF treatment correlated with a decrease in the gene expression of several pro-inflammatory cytokines, which was significant for *IL6* and *IL1B*. Likewise, the expression level of the invasive marker *MMP9* in VAT was significantly lower in anti-TNF-treated patients than in the control group (Figure 2C). Overall, these findings suggest that TNF therapies suppress tissue inflammation. Aligning with this, we found a nonsignificant trend for a decrease in NFKB-p52 protein expression in the VAT of anti-TNF-treated patients (Figure 2D).

Analysis of VAT explants corroborated that the gene expression levels of *IL6* and *IL1B* were significantly lower in anti-TNF-treated patients compared with control patients (Figure 2E), and a similar albeit nonsignificant decrease was found for *TNFA* expression. Notably, TNF-α and IL-6 secretion was also lower in VAT explants from anti-TNF-treated patients than from the control group (Figure 2F). Finally, we studied the expression levels of a panel of pro-inflammatory markers in mature adipocytes isolated from perilesional VAT, finding that only *IL34* gene expression was significantly lower in mature adipocytes isolated from anti-TNF-treated patients than from the control group (Figure 2F).

### 2.3. Anti-TNF Treatments Promote a Pro-Inflammatory-to-Anti-Inflammatory Phenotypic Switch in Adipose-Stem Cells Isolated from the Creeping Fat of Patients with Crohn’s Disease

As mentioned earlier, creeping fat hyperplasia in patients with CD is triggered by the enhanced recruitment of ASC precursors neighboring damaged intestinal areas. We previously demonstrated that ASCs from patients with CD have a bolstered pro-inflammatory and antigen-presentation capacity [21,22]. We studied whether anti-TNF treatments improved the pro-inflammatory profile in ASCs by examining the relative gene expression of typical pro-inflammatory markers (*TNFA*, *IL6*, *IL1B*, and *CCL2*) in ASCs isolated from the creeping fat of patients treated or not with anti-TNF therapy. The results of qPCR analysis clearly showed a significantly lower expression of all tested pro-inflammatory markers in ASCs isolated from anti-TNF-treated patients (Figure 3A). We next examined TNFA and IL6 secretion in 24 h conditioned medium (CM) of isolated ASCs from the two groups. Whereas no changes were observed in the concentration of TNFA between groups, the concentration of IL6 was significantly lower in the CM of ASCs isolated from patients treated with anti-TNF therapies than from the control group (Figure 3B). Contrastingly, analysis of the anti-inflammatory profile by qPCR revealed a significant increase in relative *IL10* and *TGFB1* gene expression in ASCs isolated from patients treated with anti-TNF therapies (Figure 3C). A similar trend was found for IL10 and CGSF abundance in 24 h CM of ASCs from patients treated with anti-TNF therapies in addition to a significantly greater abundance of TGB1 protein (Figure 3D). Proteomics data confirm a different secretome profiling between the ASCs isolated control group and anti-TNF group. Remarkably, IL6 was significantly lower in abundance in anti-TNF-treated CD patients compared to the control group (Figure 3E). To note, other immune-related proteins such as immunoglobulin lambda-like polypeptide 1 (IGLL1) and high mobility group nucleosomal binding domain 2 (HMGN2) were significantly lower in abundance in the anti-TNF-treated group (Figure 3E).

We used two different approaches to determine whether anti-TNF treatments can revert the immunomodulatory phenotype of ASCs from patients with CD. We first analyzed the antigen-presentation markers *HLA-DM*, *HLA-DBP, CD40*, *CD74,* and *CD80* by qPCR, observing a significantly lower expression level of *HLA-DM*, *CD74,* and *CD80* expression in ASCs isolated from anti-TNF-treated patients (Figure 3F). Then, we co-cultured ASCs of each group with and without the ovalbumin antigen and measured the relative abundance of the antigen-presentation surface protein HLA-DR and the co-stimulatory molecule CD86 by flow cytometry (Figure 3G). Under basal conditions, ASCs from anti-TNF-treated patients showed a significantly lower abundance of HLA-DR. Remarkably, in the presence of ovalbumin, significantly lower abundances of both HLA-DR and CD86 were observed in ASCs of anti-TNF-treated patients, indicating that ASCs isolated from the control group are more predisposed to trigger an inflammatory response. Moreover, ASCs from the control group had a significantly higher abundance of HLA-DR protein in the presence of ovalbumin as compared with equivalent ASCs under basal conditions (Figure 3G). Taken together, our data show that anti-TNF treatments lead to changes in the inflammatory status of ASCs of CD patients, promoting an anti-inflammatory phenotype.

## 3. Discussion

The presence of wrapping or creeping fat in perilesional regions is a distinctive feature of CD [23], and its identification is important for distinguishing the disorder from other inflammatory bowel diseases (IBDs). Remarkably, creeping fat has been found to be crucial in terms of outcomes for CD, as its elimination during surgical resection reduces the likelihood of post-operative interventions [10]. A possible explanation for this is that creeping-fat-related adipocytes intensify the inflammatory response and likely participate in the immune dysfunction observed in CD by producing pro-inflammatory adipokines that perpetuate inflammation [24]. Intriguingly, an unpublished observation made by our surgical team suggests that the morphology of the creeping fat in patients treated with biological anti-TNF therapies appears macroscopically almost normal, with less fibrosity and bleeding compared with the creeping fat obtained from untreated patients, and there is often less fat mass surrounding the damaged portion of the intestine. Studies on the effect of anti-TNF biologics on the morphology and other characteristics of creeping fat are, however, scarce. Only one study by Clemente and colleagues [25] focused on the impact of IFX on mesenteric adipose tissue in a TNBS-induced colitis rat model. The authors proposed that IFX treatment controls intestinal inflammation and modifies adipokine production in mesenteric adipose tissue, increasing the production of anti-inflammatory cytokines (IL-10 and resistin). In the present study, we observed that patients with CD on anti-TNF treatments showed suppressed gene expression and production of pro-inflammatory cytokines in the creeping fat, although we failed to observe a clear effect on anti-inflammatory cytokines (*IL10*, *TGFB1*, and adiponectin). However, we did observe a significantly higher production of anti-inflammatory cytokines in adipocyte progenitors isolated from patients treated with anti-TNF drugs than in equivalent ASCs from the control group. Of note, our proteomics results showed a decline in IL6 abundance in ASCs isolated from patients receiving anti-TNF treatment, which is consistent with a decline in IL-6 levels in the patients’ serum. We previously demonstrated that CD disturbs the immune properties of ASCs isolated from the creeping fat, which show enhanced inflammatory, phagocytic, and invasive capacities as compared with ASCs isolated from healthy individuals [21] and bolstered the antigen-presenting capacity [22]. In this study, we demonstrate that anti-TNF therapies revert these immunomodulatory capacities in the ASCs isolated from anti-TNF-treated patients as we observed lower gene expression levels of antigen-presentation markers and co-stimulatory molecules in these cells. In agreement with those findings, the outcome of our proteomics analysis revealed that IGLL1 and HMGN2, two immune-response-related proteins [26,27], were less abundant in the ASCs of CD patients that received anti-TNF therapy.

IL34 is up-regulated in the mucosa of patients with IBD, and a role for this cytokine in sustaining the inflammatory responses in IBD has been suggested [28]. It has been described that in response to TNFA, IL34 acts as a crosstalk molecule between ASCs and the immune cells in the inflamed gut, contributing to the proinflammatory response mediated by the inflammatory cytokines secreted by the immune cells infiltrated in this tissue [29]. Neutralization of TNF-alpha with IFX was found to reduce IL34 production in the inflamed gut mucosa of patients with CD [28]. We observed a reduction in *IL34* gene expression also in mature adipocytes isolated from the creeping fat of patients with CD treated with anti-TNF drugs, indicating that anti-TNF therapies could regulate IL34 production in inflamed gut mucosa and creeping fat. Taking into account all this evidence, we think that IL34 might be a potential therapeutic target to ameliorate the proinflammatory ambient in the intestine of IBD patients. Nevertheless, it has also been reported that IL34 plays a key role in the recruitment and differentiation of resident macrophages that contribute to the maintenance of the intestinal epithelium by secreting IL10 anti-inflammatory cytokines [30]. Because of that, further studies will be needed to better ascertain IL34 functions to better interpret its role in IBD.

Our results indicate that anti-TNF therapies re-establish the adipose tissue architecture, restoring the adipocyte size to that of healthy VAT adipocytes. Additionally, we found that anti-TNF treatments diminished the infiltration of macrophages and T lymphocytes in VAT. Here, we demonstrate that anti-TNF therapies normalize the phenotype of ASCs, indicating that these drugs might have beneficial effects on CD by improving inflammation and restoring the morphology of the creeping fat. Along this line, anti-TNF drugs might act on the VAT by blocking the production of TNF-α in this inflamed tissue. This favorable effect might, however, have harmful consequences, as the VAT could partly retain the drugs, acting as a sink and preventing them from reaching the intestinal mucosa. While further studies are necessary to test this hypothesis, it might explain the ultimate loss of response to anti-TNF drugs observed in patients with CD (known as secondary nonresponders) [31]. Accordingly, TNF-α and anti-TNF drug levels in intestinal mucosa and the creeping fat could be measured in primary and secondary nonresponders in future cohorts. Nevertheless, our results indicate that anti-TNF treatment acts at the level of the creeping fat, changing its morphology and pro-inflammatory status. Increasing the doses of the anti-TNF drug according to the abundance of VAT or using the creeping fat index [4] may be interesting options to apply in clinical procedures. According to some authors, BMI is a reliable predictor of anti-TNF medication failure in patients with ulcerative colitis [32]. However, studies on patients with CD are contradictory and inconsistent [33]. Although it is commonly known that patients with CD have a greater abundance of VAT than healthy controls [34], this cannot be accurately measured by simple nonspecific anthropometric measures such as BMI, which only considers weight and height.

The fact that IFX treatment (or other biologics) is associated with weight gain is still controversial. There are some authors that affirmed that weight gain is not associated with IFX treatment in IBD patients [35], while other authors [36] observed only weight gain in a group of patients treated with IFX associated with clinical parameters such as male gender, high CRP, and low levels of albumin. In contrast, Lepp and colleagues [37] found that IFX caused weight gain in 60% of all the CD patients studied. Furthermore, the weight increment was associated with improvements in inflammatory markers and disease activity. In this sense, the authors proposed that the causes of weight gain may be related to treatment-induced metabolic changes and a reduced inflammatory burden. Intriguingly, these are the main results of our study. Anti-TNF treatment produces metabolic changes in adipose tissue architecture, increasing adipocytes size and reducing immune cells infiltration and adipose tissue inflammation. However, more research will be required to conclusively link the weight gain to the expansion of adipose tissue brought on by IFX or other anti-TNF therapies.

We believe that the abundance of VAT in each patient is likely a key factor to consider when predicting the effectiveness of anti-TNF treatment in patients with CD.

Our study has some limitations that warrant discussion. Less than 10.3% of the control group patients (n = 3 from 29) were treated with anti-TNF therapies more than ten years ago before surgical intervention. Although this fact could represent a possible bias in our study, we consider that the effects of anti-TNF drugs in these patients might be negligible regarding the long period without receiving any anti-TNF therapy. Another limitation of the study is the fact that it is an observational study with a limited number of cases, and it would be necessary to monitor the temporal evolution of the patients to evaluate changes in the VAT during anti-TNF treatment. To ascertain that, larger prospective studies will be necessary to decipher the clinical relevance of these findings. In addition, increased VAT has been previously linked to changes in the gut microbiota [38], with similar alterations found in CD [39,40]. The interaction between dysbiotic gut microbiota and the host immune system is another potential mechanism by which visceral obesity is linked to CD recurrence. If anti-TNF therapies suppress inflammation and immune cell infiltration in the VAT of patients with CD, this may have potential implications for the medical and postsurgical management of CD, which merits study. Importantly, our data indicate that anti-TNF therapies might help control intestinal inflammation, improving the features of perilesional adipose tissue in patients with CD.

## 4. Materials and Methods

### 4.1. Study Design

Patients with CD treated with anti-TNF therapies and those nontreated with anti-TNF treatment (control group) who met the inclusion criteria were recruited from the outpatient Surgery Clinic of the University Hospital Joan XXIII (Tarragona) and University Hospital Vall d’Hebron (Barcelona). The study was conducted according to the tenets of the Helsinki Declaration and was approved by the ethics committees of each hospital (references CEIM 177/2018 and PR[CS]383/2021, respectively). All participants signed an informed consent. Samples of VAT near the inflamed intestinal region were aseptically collected from patients who were treated with anti-TNF antibodies (IFX or ADA) (n = 14) (59% ileum, 17% colon, and 24% ileocolon) and from patients who never received any biological treatment (n = 29) (43% ileum, 28.5% colon, and 28.5% ileocolon). VAT biopsies were obtained during a scheduled surgery for symptomatic complications. Tissue samples were aseptically collected, and all were obtained before bowel opening or resection to avoid possible contamination. Clinical data, anthropometric, demographic, and biochemical variables of the cohort are shown in Table 1.

### 4.2. Histological Studies

#### 4.2.1. Haematoxylin and Eosin Staining

VAT samples (~100 mg) were washed with PBS and fixed in 4% paraformaldehyde (Sigma-Aldrich, Madrid, Spain) overnight at 4 °C. Samples were then dehydrated and degreased before paraffin embedding, and tissue sections (2 µm thickness) were stained with H&E. SlideViewer software was used for visualization of stained adipose tissue [41,42]. Five circular areas of 304,558 µm^2^ were randomly selected from each sample. One representative image of each circular section was captured at 20×, allowing for an overview of ~250 adipocytes per field. Each image was analyzed using the Adiposoft plug-in of Fiji software (ImageJ, NIH), setting a minimum equivalent diameter (D-eq) of 35 µm and a maximum of 300 µm [19].

#### 4.2.2. CD68 and Double CD68-CD3 Immunohistochemistry

For immunohistochemistry analysis, tissue sections (4 µm) were heated at 96 °C for 20 min for antigen retrieval and then incubated for 20 min with a primary antibody against CD68 (Clone KP1, ready-to-use, cat IR609; Agilent/Dako, Tokyo, Japan). For double labelling of CD68/CD3 antigens, tissue sections were first stained with the CD68 antibody and were then incubated with a primary antibody against CD3 (Clone 56C6, ready-to-use, cat IR648; Agilent/Dako) [43]. The END-VISION^TM^ FLEX visualization system (Agilent/DAKO, Santa Clara, CA, USA) was used with 3,3′-diaminobenzidine chromogen as the substrate, followed by counterstaining with haematoxylin. Immunostainings were performed in an Autostainer Link 48 instrument (Agilent/DAKO). SlideViewer software was used for visualization of CD68- and CD68/CD3-stained adipose tissue. Macrophages (CD68-positive) were magenta-stained and T cells (CD3-positive) were brown-stained. In total, 10 circular sections of 304,558 µm^2^ were randomly selected from each sample and images were taken at 20×. Macrophages and T cells were counted in single sections excluding those found in or adjacent to a blood vessel. Thus, only truly infiltrated immune cells were considered. Misleading stained cells, which included both colors, were also dismissed.

### 4.3. Isolation and Culture of Adipose Tissue-Derived Stem Cells

ASCs were isolated as described [44,45]. Briefly, VAT biopsies were washed extensively with PBS (Cultek SL, Madrid, Spain) to remove debris, and were then treated with 2 mg/mL of collagenase type I (Sigma-Aldrich) in PBS-1% bovine serum albumin (Sigma-Aldrich) at 37 °C for 1 h with gentle agitation. Digested samples were centrifuged at 300× *g* at 4 °C for 5 min to separate adipocytes from stromal cells. Thereafter, mature adipocytes were collected, washed with sterile PBS, and frozen in liquid nitrogen for storage at −80 °C until required. The cell pellet containing the stromal vascular fraction (SVF) was re-suspended in stromal culture medium (DG) consisting of Dulbecco’s modified Eagle’s medium high-glucose/Ham’s F12 medium (1:1) with 10% fetal bovine serum (Sigma-Aldrich) and 1% antibiotic/antimycotic (penicillin, streptomycin, and amphotericin). The SVF suspension was seeded in a T25 flask and incubated in a humidified incubator with 5% CO_2_, 95% O_2_ at 37 °C. The medium was replaced 24 h after seeding to remove nonadherent cells and was replenished every 2–3 days thereafter to ensure optimal nutrient supplementation. Primary ASCs cultures at passage 0 (P0) were grown to 90% confluence to prevent spontaneous differentiation and were detached with trypsin-EDTA (ThermoFisher Scientific, Waltham, MA, USA). Aliquots (1 × 10^6^) were cryopreserved at −80 °C until required [46]. All experiments were performed in cells at P3–4. Adipose tissue (VAT)-derived macrophages (ATMs) were also isolated from the SVF of biopsies, as described [47,48,49,50].

### 4.4. Adipose Stem Cell Immunophenotyping

ASCs (2 × 10^5^) were incubated with a panel of primary antibodies (described in Appendix A). After isolation, the minimal functional and quantitative criteria, established by the International Society of Cell Therapy and the International Federation for Adipose Therapeutics and Science, were confirmed by flow cytometry using the FacsAriaIII (BD, Biosciences, San Jose, CA, USA). Data analysis was performed using the FACSDiva software (BD Biosciences, San Jose, CA, USA) [22,51].

### 4.5. Isolation and Incubation of Adipose Tissue Explants

Fresh explants of each VAT biopsy (~200 mg) were placed in 1 mL of DG medium. Samples were incubated for 24 h in 5% CO_2_, 95% O_2_ at 37 °C. Finally, VAT explants and 24 h CM were frozen in liquid nitrogen and cryopreserved at −80 °C until required.

### 4.6. RNA Extraction and Real-Time Quantitative PCR

RNA was extracted from ~200 mg of VAT (total VAT and from explants) and from 100,000 VAT-derived ASCs, mature adipocytes, and ATMs using the TriPure Isolation Reagent (Roche, Basel, Switzerland). RNA concentration was determined by absorbance at 260 nm, and purity was estimated using a NanoDrop spectrophotometer (NanoDrop Technologies Inc., Wilmington, DE, USA). cDNA was synthesized using SuperScript II reverse transcriptase and random hexamer primers (Invitrogen Life Technologies, Darmstadt, Germany). The purity of each extraction was determined by the OD_260_/OD_280_ ratio. Gene expression was studied by using real-time (q)PCR on a Quant Studio 7 Pro instrument (Applied Biosystems, Waltham, MA, USA) and was calculated using the comparative CT method (2^−ΔΔCT^) and normalized to the expression of the housekeeping gene *18S* (TaqMan probes listed in Appendix A).

### 4.7. Western Blotting

VAT was lysed and homogenized in M-PER buffer containing a Protease Inhibitor Cocktail, and protein concentrations were determined with the BCA Protein Assay Kit (all from Pierce Biotechnology Inc., Rockford, IL, USA). Equal amounts of total protein were separated by SDS-polyacrylamide gel electrophoresis, transferred to Immobilon membranes (Millipore, Billerica, MA, USA), and blocked. Membranes were probed with polyclonal antibodies against CD14 (Cell Signaling Technology, Danvers, MA, USA), NFKB (Santa Cruz Biotechnology, Santa Cruz, CA, USA), and FFA (Santa Cruz Biotechnology), which was used as a loading control. Immunoreactive bands were visualized with the SuperSignal West Femto chemiluminescent substrate (Pierce Biotechnology), and images were captured using the VersaDoc imaging system and Quantity One software (BioRad, Hercules, CA, USA).

### 4.8. Cytokine Secretion

Twenty-four-hour CM from VAT explants and ASCs as well as serum from patients were used to determine the levels of the inflammatory cytokines TNFA and IL6 by enzyme-linked immunoassay (ELISA) (R&D systems, DTA00D and HS600C, respectively).

### 4.9. Proteomics Analysis of Conditioned Medium from Adipose-Derived Stem Cells

We used an untargeted proteomics approach consisting of liquid chromatography coupled to tandem mass spectrometry to compare and identify differences in protein levels secreted into the extracellular (conditioned) medium by ASCs isolated from the creeping fat of the control group (n = 6) and anti-TNF group (n = 3). We reanalyzed a subcohort of our proteomic data [50] looking for differences in protein abundance between ASCs isolated from anti-TNF-treated and -untreated patients.

#### Data Analysis

Acquired spectra were analyzed using the Proteome Discoverer software suite (v2.3, Thermo Fisher Scientific) and the Mascot search engine (v2.6, Matrix Science Ltd., Boston, MA, USA), as described [50,52]. Proteomic data were evaluated to identify significant differences (*p* < 0.05) in proteins between both groups using the DEP package that provides an integrated analysis workflow for robust and reproducible analysis of mass spectrometry proteomics data [53]. Briefly, proteins that were not present in 3 of the 6 samples for the control group and 2 of the 3 samples for the anti-TNF group were not considered for statistical analysis, resulting in 614 proteins retained. Filtered data were then background-corrected and normalized by variance stabilizing transformation [54]. Protein-wise linear models combined with empirical Bayes statistics were used for the differential enrichment analysis using the limma package [55]. All statistical data treatments were performed with the R statistical programming environment (version 4.1.3; [56]). The mass spectrometry proteomics data have been deposited in ProteomeXchange with the accession code PXD030520.

### 4.10. Antigen-Presentation Assay

A total of 6 × 10^4^ ASCs were co-cultured or not with 5 μg/mL of ovalbumin (Alexa Fluor488 cat 034781) and incubated overnight at 37 °C and 5% CO_2_. Subsequently, cells were incubated for 30 min at 4 °C with a PE-conjugated monoclonal antibody against CD86 and a Pe-Cy7-conjugated antibody against HLA-DR (BD Pharmingen, Erembodegem, Belgium), and were washed twice. During flow cytometry, dead cells were identified using propidium iodide and were gated out by setting a threshold on the forward scatter. Ten thousand cells were acquired for each exposure condition and were analyzed on a FACStar Plus flow cytometer using CellQuest software (Becton Dickinson, Sunnyvale, CA, USA) [22]. The results of the phenotypic analysis were expressed as percentages of positive cells.

## 5. Conclusions

We found that anti-TNF therapies restore adipocyte morphology and suppress inflammation in the creeping fat of patients with CD at least in part by switching the phenotype of adipocyte precursors and VAT from pro- to anti-inflammatory. We propose that it might be relevant to consider VAT, specifically creeping fat, when anti-TNF medications are used in patients with CD.

## Figures and Tables

**Figure 1 ijms-23-11170-f001:**
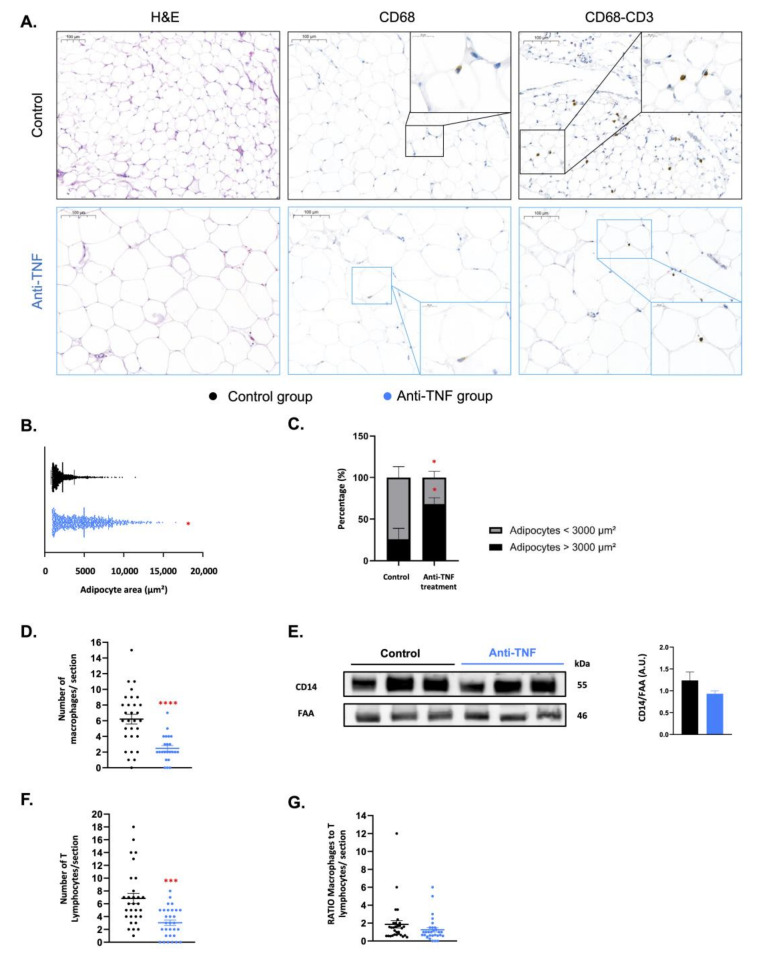
Anti-TNF therapy remodels adipose tissue, restoring normal adipocyte size and suppressing immune cell infiltration. (**A**) Representative histological images obtained from randomly selected 304,558-µm^2^ circular sections. Left panel: hematoxylin & eosin (H&E) staining. Central panel: CD68 immunohistochemistry (brown staining). Right panel: double CD68 (magenta stain) and CD3 immunohistochemistry (brown stain). (**B**) Adipocyte area in the creeping fat of control group and anti-TNF-treated group. (**C**) Percentage of large-size (>3000 µm^2^) and small-size (<3000 µm^2^) adipocytes detected in creeping fat. (**D**) Number of CD68-positive macrophages per section detected in creeping fat. (**E**) CD14 protein expression analyzed by Western blotting in creeping fat of control and anti-TNF-treated group. (**F**) Number of CD3-positive T lymphocytes in the creeping fat of control and anti-TNF-treated group. (**G**) Ratio of CD68- to CD3-positive cells in creeping fat. Results are shown as mean ± SEM from independent donors’ experiments. * *p* < 0.05, *** *p* < 0.001, and **** *p* < 0.0001 vs. control group.

**Figure 2 ijms-23-11170-f002:**
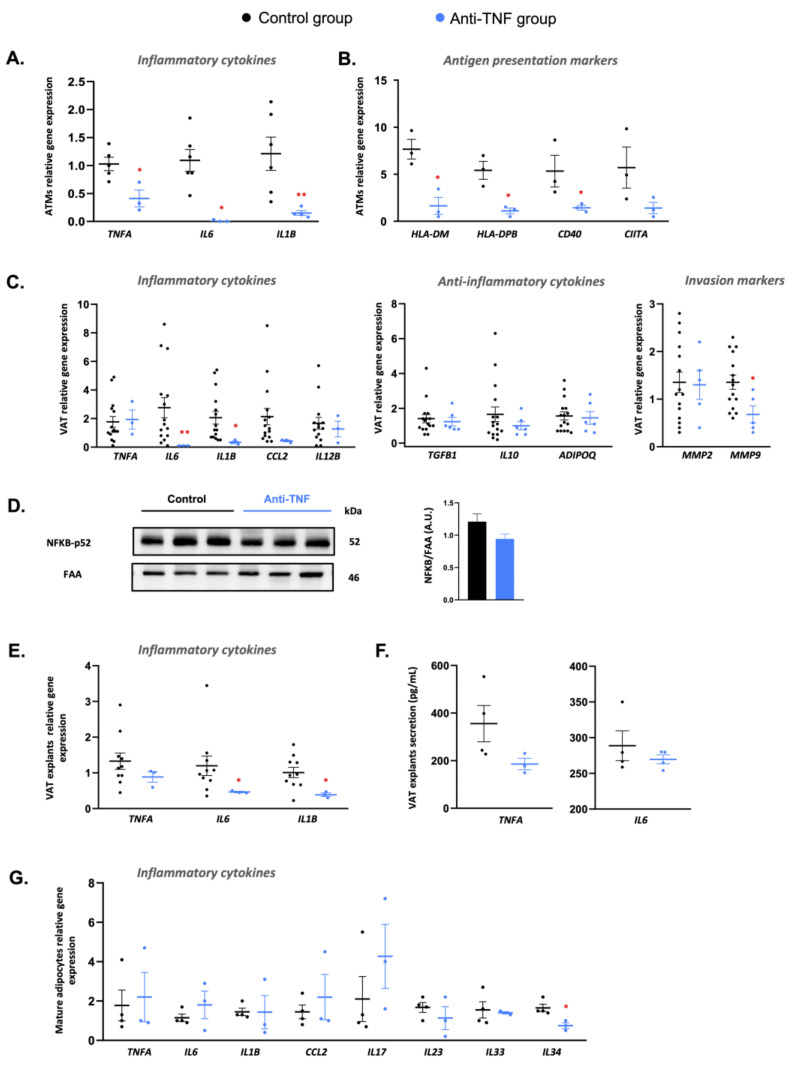
Anti-TNF treatment suppresses inflammation in different cell subtypes of creeping fat. (**A**) Gene expression of pro-inflammatory genes (*TNFA*, *IL6*, and *IL1B*) in VAT-derived adipose tissue macrophages (ATMs). (**B**) Gene expression of antigen-presentation markers in VAT-derived ATMs. (**C**) Gene expression of the inflammatory cytokines *TNFA, IL6*, *IL1B, CCL2,* and *IL12B;* anti-inflammatory cytokines *TGFB1*, *IL10*, and *ADIPOQ*; the invasion markers *MMP2* and *MMP9* in the creeping fat of patients treated with anti-TNF or control group. (**D**) NFKB-p52 protein expression analyzed by Western blotting in the creeping fat of patients treated with anti-TNF or control group (**E**) Gene expression of the pro-inflammatory cytokines *TNFA*, *IL6*, and IL1B in VAT explants isolated from patients treated with anti-TNF or control group. (**F**) TNFA and IL6 abundance in 24 h conditioned medium of VAT explants from patients treated with anti-TNF or control group. (**G**) Inflammatory cytokine gene expression in mature adipocytes isolated from patients treated with anti-TNF or control group. Results are shown as mean ± SEM from independent donors’ experiments. * *p* < 0.05 and ** *p* < 0.01 vs. control group.

**Figure 3 ijms-23-11170-f003:**
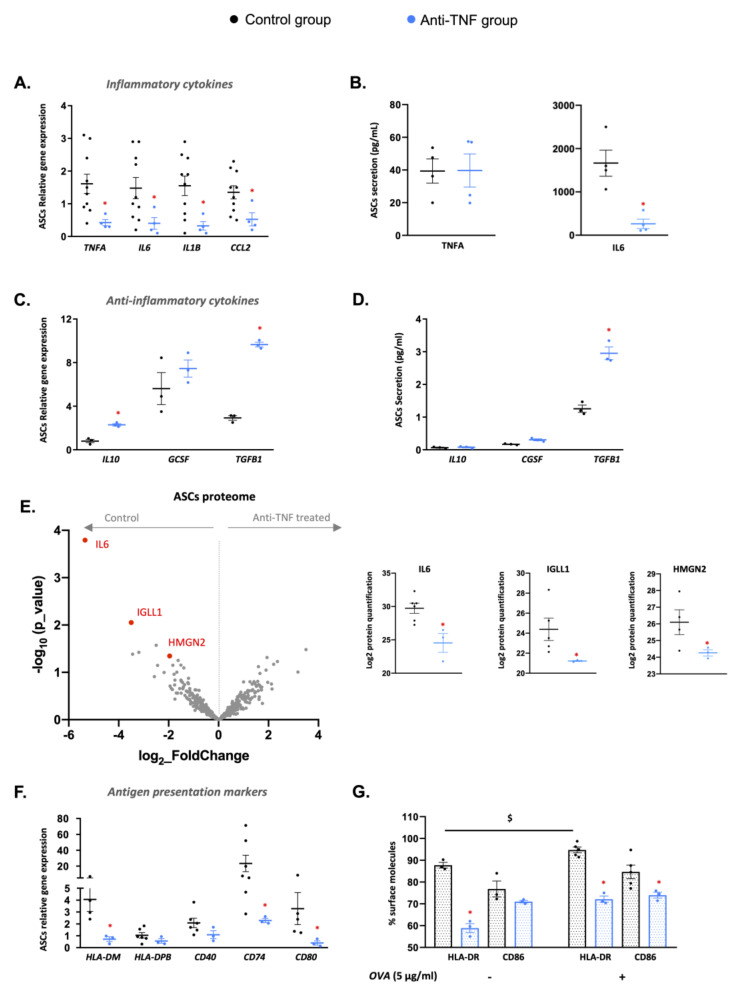
Anti-TNF therapies modify the inflammatory status of creeping fat-derived ASCs. (**A**) Gene expression of pro-inflammatory genes (*TNFA*, *IL6*, *IL1B*, and *CCL2*) in adipose-stem cells (ASCs) and (**B**) TNFA and IL6 abundance in 24 h conditioned medium of ASCs from patients treated with anti-TNF or control group. (**C**) Gene expression of anti-inflammatory genes (*IL10*, *CGSF*, and *TGFB1*) in ASCs from patients treated with anti-TNF or control group. (**D**) TGFB1 protein abundance in 24 h conditioned medium of ASCs from patients treated with anti-TNF or control group. (**E**) Analysis of the ASC protein secretome and protein abundance differences between ASCs isolated from control group (n = 6) and anti-TNF-treated group (n = 3). (**F**) Gene expression of antigen-presentation markers (*HLA-DM* and *HLA-DPB*) and co-stimulatory molecules (*CD40*, *CD74*, and *CD80*) in ASCs from patients treated with anti-TNF or control group. (**G**) ASCs isolated from patients treated with anti-TNF or control group co-cultured or not with ovalbumin (OVA) were stained with a panel of antibodies and analyzed by flow cytometry to compare the percentage of surface expression of antigen-presenting markers. Results are shown as mean ± SEM from independent donors’ experiments. * *p* < 0.05 vs. control group. $ *p* < 0.05 control group without OVA vs. control group with OVA.

**Table 1 ijms-23-11170-t001:** Demographic characteristics and clinical data of patients.

	Control Group	Anti-TNF Group
**N**	29	14
**Sex (male/female)**	14/15	5/7
**Age**	40.67 ± 12.05	46.4 ± 14.69
**BMI (kg/m^2^)**	23.48 ± 3.98	24.80 ± 7.4
**Smoking status, n (%)**		
Current smoker	8 (28)	5 (36)
Never smoker	19 (66)	8 (57)
Ex-smoker	2 (7)	1 (7)
**Age at diagnosis, n (%)**		
A1	1 (5)	2 (14)
A2	17 (81)	8 (57)
A3	3 (14)	4 (29)
**Location, n (%)**		
L1	17 (59)	6 (43)
L2	5 (17)	4 (29)
L3	7 (24)	4 (29)
**Behavior, n (%)**		
B1	5 (17)	4 (29)
B2	14 (48)	7 (50)
B3	10 (34)	3 (21)
**Previous surgery**	5 (17)	24 (83)
**Indications for surgery (Stenosis/Abscess/Inflammation), n**	(20/8/1)	(13/1/0)
**Corticoid treatment, n (%)**	8 (28)	5 (36)
**Anti-TNF treatment** (adalimumab/infliximab)	-	9/5
**Duration anti-TNF treatment**, in months	-	13 ± 4.6
**Time last dose before surgery**, in days	-	19.7 ± 2.8
**Other anti-TNF received, n (%)**	-	2 (21)
**Concomitant immunosuppressors (Thiopurines), n (%)**	-	6 (43)
**C-reactive protein (mg/dL)**	4.66 ± 1.10	1.88 ± 1.48 ^a^
**Serum levels IL-6 (pg/mL)**	14.86 ± 4.19	4.36 ± 1.67 ^a^

Abbreviations: BMI, body mass index; Age at diagnosis: A1 ≤ 16 years; A2 17–40 years; A3 > 40 years; Location: L1 = ileal; L2 = colonic; L3 = ileocolonic; Behavior: B1 = non-stenotic, non-fistulizing Crohn’s disease; B2 = stenotic Crohn’s disease; B3 = fistulizing Crohn’s disease. ^a^ *p* < 0.05 significant differences compared with control group.

## Data Availability

Any data or material that support the findings of this study can be made available by the corresponding author upon request.

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
