# Peer review of "Anti-TNF Therapies Suppress Adipose Tissue Inflammation in Crohn’s Disease"

_ijms, 2022, doi:10.3390/ijms231911170_

Round 1

Reviewer 1 Report

A necessary study to confirm aspects of the adipose tissue of patients with CD that we all suspected but that we had no evidence for. 

There are only 2 aspects that could clarify the results generated and that I invite the authors to discuss. On the one hand, it would be very interesting to know the number of Anti-TNF treatments that patients have received; this could indicate if there are cumulative effects on adipose tissue and immune system. On the other hand, knowing the long-term toxicity of repetitive anti-TNF treatments and its effect on white fat, and changes in visceral fat volume, it would be interesting to discuss the risks of future surgeries in patients with anti-TNF treatment.

Line 359, modificate by: After isolation, some of the membrane markers recommended by ISCT and IFATS were confirmed by flow cytometry ... . Both Societies recommend a higher panel of markers as well as an analysis of differentiation capacity.

Reviewer 2 Report

Patients were operated because of “symptomatic complications” (line 305); in order to judge the findings presented in the paper we need to know, a) what kind of complications (strictures, fistulae, abscesses, dysplasia?), b) continuing ongoing inflammation despite treatment, c) if these complications were evenly distributed in the two groups, and, d) if any of the patients in the non-aTNF-group previously had been treated with aTNFs (possible bias). If there e.g. is an excess number of patients operated on for ‘inflammation’ in the aTNF-group compared to indolent strictures in the non-aTNF-group, the results need to be interpreted very cautiously.

The authors should provide information on: i) Treatment duration in months and dosing of aTNF in patients, ii) The time elapsed between the last dose of aTNF and surgery, iii) Any concomitant medication except for corticosteroids, iv) Number of patients that had their FIRST or SUBSEQUENT intestinal resection.

Weight increases in CD-patients under treatment with IFX (and other biologics) is well-known and should be discussed. See: Lepp J et al, Scand J Gastroenterol. 2020 Dec;55(12):1411-1418.; Kaazan P, et al Dig Dis Sci. 2022 Apr 3. doi: 10.1007/s10620-022-07488-7; Winter RW, et al. Am J Gastroenterol. 2022 May 1;117(5):777-784. Is this weight increase related to a general expansion of adipose tissue (cellular hyperplasia as seen in Fig 1A?)?

Discussion: the authors present no clear evidence for their statements on the macroscopic appearance of VAT with and without aTNF-treatment (lines 75-77 in Results) and lines 223-228 in Discussion. Please expand and provide more information; state at least ‘unpublished observation’ or ‘personal communication’.

For the discussion on IL-34 (lines 250-251) the authors should reference and comment two additional papers in the field: Zwicker S, et al. Clin Sci (Lond). 2015 Aug;129(3):281-90; Monteleone G, et al. Front Immunol. 2022 Apr 22;13:873332. doi: 10.3389/fimmu.2022.873332.

Minor

The text can be slightly shortened, e.g by renaming the two groups as 'anti-TNF-group' and 'control group'.

Were any results different in patients treated with ADA (n=9) versus IFX (n=5)?

Were there any correlation between any result and the patient’s BMI?

Table 1: give percentages without decimals. Line 108: rephrase to …”anti-TNF naïve group”.

Figure 2: Expand the figure text 2C and 2E to include all cytokines in the respective graphs.

IL-6 was measured in serum (line 240) but no results are provided; omit or include results.
